# Rapid Earthquake Damage Assessment System in the Black Sea Basin: Selection/Adoption of Ground Motion Prediction Equations with Emphasis in the Cross-Border Areas

Nikolaos Theodoulidis [1,*], Basil Margaris [1], Dimitris Sotiriadis [2], Can Zulfikar [3], Seyhan Okuyan Akcan [4], Carmen Ortanza Cioflan [5,6], Elena Florinela Manea [5,6] and Dragos Toma-Danila [5,6]

[1] Institute of Engineering Seismology and Earthquake Engineering, 55535 Thessaloniki, Greece; margaris@itsak.gr

[2] Department of Civil Engineering, Democritus University, 67100 Xanthi, Greece; dsotiria@civil.duth.gr

[3] Disaster Management Institute, Istanbul Technical University, Istanbul 34469, Turkey; aczulfikar@gtu.edu.tr

[4] Civil Engineering Department, Bogazici University, Istanbul 34342, Turkey; seyhan.okuyan@boun.edu.tr

[5] National Institute for Earth Physics, 077125 Magurele, Romania; cioflan@infp.ro (C.O.C.); elena.manea@infp.ro (E.F.M.); toma@infp.ro (D.T.-D.)

[6] Civil Engineering Department, Ovidius University, 900527 Constanța, Romania

* Correspondence: ntheo@itsak.gr

**Abstract:** In the present study, an effort to propose and adopt appropriate Ground Motion Prediction Equations (GMPEs) for the Rapid Earthquake Damage Assessment System (REDAS) in the Black Sea basin is attempted. Emphasis of GMPE harmonization in the cross-border areas (CBA) is given. For this reason, two distinct sub-areas are investigated, taking into consideration their seismotectonic regime. One sub-area refers to active shallow crustal earthquakes (Greece-Turkey, CBA) and the other to intermediate-depth and shallow crustal earthquakes (Romania-Moldova, Western Black Sea CBA). Testing and ranking of pre-selected GMPEs has been performed using strong motion data of the broader CBA regions of both sub-areas. The final proposed GMPEs to feed the REDA System may assure the effective estimation of ShakeMaps and—in combination with the appropriate vulnerability curves—reliable near-real-time damage assessment in the cross-border earthquake affected areas.

**Keywords:** ShakeMaps; rapid earthquake damage assessment; GMPEs; data harmonization; seismologic services

## 1. Introduction

Near-real-time estimation of the ground motion induced by an earthquake (ShakeMap) is based on the combination of recorded ground intensity together with that computed using Ground Motion Prediction Equations (GMPEs). The selection of appropriate GMPEs is an essential step in defining seismic input for the Rapid Earthquake Damage Assessment System (REDAS) (Papatheodorou et al., 2023) [1]. During the past half-century, a plethora of strong motion attenuation relations have been proposed worldwide, known as Ground Motion Prediction Equations (GMPEs) or Ground Motion Predictive Models (GMPMs) (see, among others, Douglas (2021) [2]). These models are extremely useful tools in seismic hazard assessment and more recently in ShakeMaps generation. They are based on earthquake strong motion recordings from various worldwide seismotectonic regimes, considering effects due to local or/and regional seismotectonic regimes as well. Although the definition of GMPEs must not be related to countries' borders, in many cases strong motion data acquired by country-level networks is preferred for determining local GMPEs as the most representative in seismic hazard assessment for a specific country. Consequently, in the cross-border areas of certain countries, the "paradox" of differing expected ground motion intensity is observed, depending on the GMPEs used by each country.

To tackle this "paradox" in the cross-border areas (CBAs) of the broader Black Basin territory, the adoption of the most appropriate GMPEs is attempted, thus providing reliable input to the Rapid Earthquake Damage Assessment System (REDAS). For the purposes of REDA, the broader Black Sea Basin territory is defined in Figure 1. In fact, all countries around the Black Sea are included in this territory. However, in the framework of the REDACt project, only four countries participated (Romania, Moldova, Greece, Turkey). Consequently, there are two distinct cross-border areas: (a) the Greece–Turkey CBA and (b) the Romania–Moldova CBA. It must be stated that the structure/architecture of the REDA System can in future include additional countries, provided they contribute to the System with harmonized data and information.

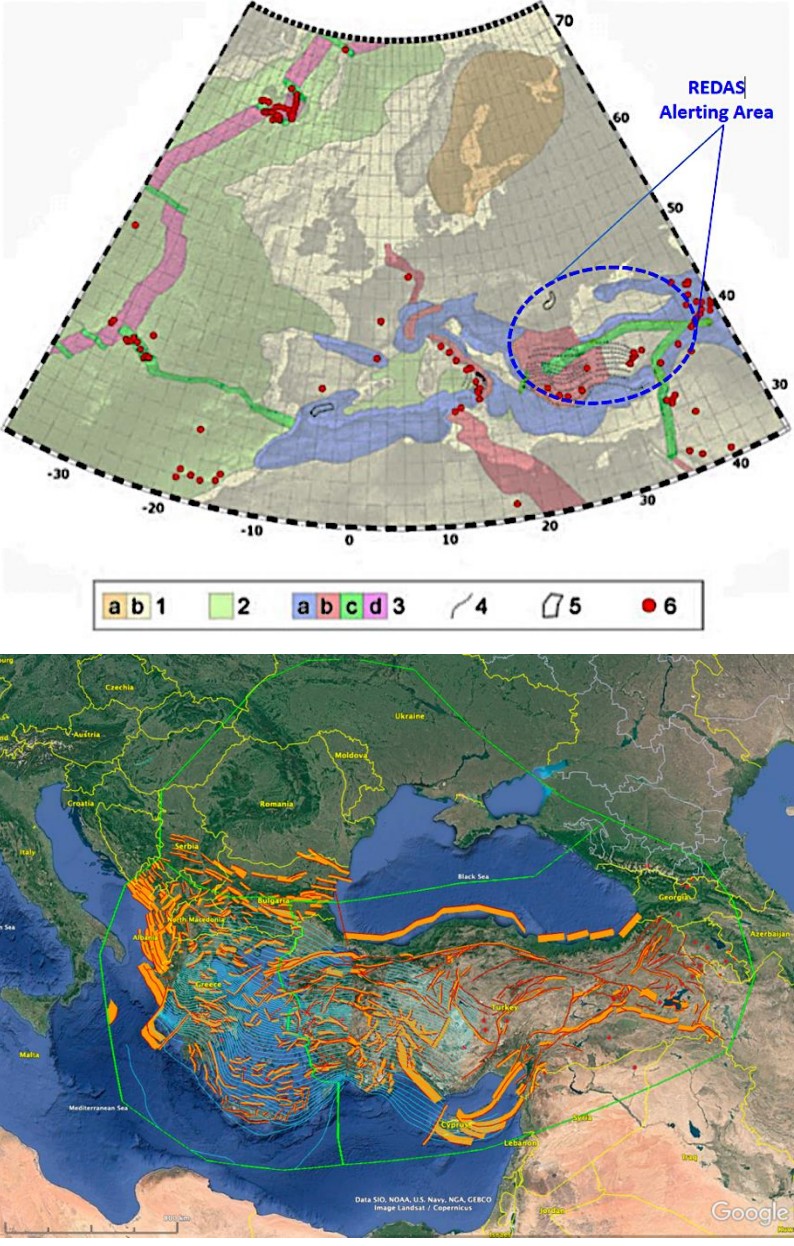

**Figure 1.** (**top**) Seismotectonic map of the Euro-Mediterranean area developed for the SHARE prj. (modified from Delavaud et al., 2012 [3]; Woessner et al., 2015 [4]). 1. SCR, shield (a) and continental crust (b); 2. oceanic crust; 3. ASCR, compression-dominated areas (a) including thrust or reverse faulting, and contractional structures in the upper plate of subduction zones; extension-dominated areas (b) including associated transcurrent faulting; major strike-slip faults and transforms (c), and

mid oceanic ridges (d); 4. subduction zones shown by contours at 50 km depth interval of the dipping slab; 5. areas of deep-focus non-subduction earthquakes; 6. active volcanoes and other thermal/magmatic features. (**bottom**) Zoom in the REDAS Alerting Area: Border lines (yellow) of the study Alerting Area (light green) along with the seismic sources, seismogenic faults, subduction zone isodepths, and volcanoes/magmatic features (SHARE prj. 2013; https://www.seismofaults.eu/services/efsm20-services, accessed on 28 February 2024).

According to the seismotectonic map of the Euro-Mediterranean, which has been developed in the framework of the SHARE project (https://www.seismofaults.eu/services/efsm20-services, accessed on 28 February 2024) (Figure 1), the eligible earthquake Alerting Area to be included in the REDAS, in terms of tectonic features, can be divided into two general categories: (i) one including the Active Shallow Crustal Regions (ASCRs) of Greece and Turkey, and (ii) another Stable Continental Region (SCR) consisting mainly of continental crust in the Romania, Moldova, and the Black Sea area. For this reason, two parallel investigations regarding the selection and ranking of the most appropriate GMPEs are presented below.

## 2. Selection and Ranking of GMPEs for CBAs of Greece and Turkey

A set of 37 GMPEs based on data from the broader area of Greece and Turkey that were published in international peer-reviewed journals for the period 1985–2021 (Douglas 2021) [2] have been compiled [5]. Most recently published GMPEs generally utilize advanced independent parameters in regression analyses (e.g., higher-quality digital data, faulting type, the site amplification $V_{S30}$ proxy) and they are based on data already incorporated in older GMPEs. For this reason, it was decided to restrict our analyses only to GMPEs published during the last decade. The selected 7 out of 37 GMPEs for the period 2013–2021 are shown in Table 1. The rest of them were excluded based on the criteria mentioned above as well as on those suggested by Cotton et al. (2006) [6] and Bommer et al. (2010) [7], as summarized below: the GMPE is from a clearly irrelevant tectonic regime; it is not published in an international peer-reviewed journal; the documentation of the GMPE and its underlying dataset is insufficient; it is superseded by more recent publications; its frequency range is not appropriate for engineering applications; it has an inappropriate functional form; or the regression method is judged to be inappropriate. Subsequently, evaluation and ranking of the predictive performance of the selected GMPEs against strong motion data for the CBAs of Greece and Turkey was attempted.

**Table 1.** Selected for ranking GMPEs of the Greece and Turkey CBAs.

| GMPE | Magnitude Type/Range | Distance Type/Range | Intensity Measure | Site Classification Type | Faulting Style [*] | Horizontal Component Type | Region |
|---|---|---|---|---|---|---|---|
| Akkar et al. (2014) [8] | $M_w$/4.0–7.6 | $R_{jb}$, $R_{hypo}$ or $R_{epi}$/1–200 km | PGA, PGV, $S_a$ (T = 0.02–4.0 s) | $V_{S30}$-based | NS, SS, RS | Geometric Mean | Europe and Middle East |
| Chiou and Youngs (2014) [9] | $M_w$/3.5–8.5 for SS $M_w$/3.5–8.5 for NS or RS | $R_{rup}$, $R_{jb}$,$R_x$/0–300 km | PGA, PGV, $S_a$ (T = 0.01–10.0 s) | $V_{S30}$-based (180–1500 m/s) | NS, SS, RS | Arithmetic mean | California, Japan, China, Italy, Turkey |
| Abrahamson et al. (2014) [10] | $M_w$/3.0–8.5 | $R_{rup}$, $R_{jb}$, $R_x$, $R_{y0}$/0–300 km | PGA, PGV, $S_a$ (T = 0.01–10.0 s) | $V_{S30}$-based | NS, SS, RS | Arithmetic mean | California, Japan, China, Taiwan Italy, Turkey |
| Chousianitis et al. (2018) [11] | $M_w$/4.0–6.8 | $R_{epi}$/ 0.3–200 km | PGA, PGV, $T_m$ | NEHRP classification (B, C, D) | Unknown, NS, SS, RS | Geometric Mean | Greece |
| Kotha et al. (2020) [12] | $M_w$/3.0–7.4 | $R_{jb}$/1–545 km | PGA, PGV, $S_a$ (T = 0.01–8.0 s) | $V_{S30}$-based (90–3000 m/s) or slope-based | - | RotD50 | Europe and Mediterranean |
| Boore et al. (2021) with bias [13] | $M_w$/4.0–8.0 | $R_{jb}$/1–300 km | PGA, PGV, $S_a$ (T = 0.01–10.0 s) | $V_{S30}$-based (150–1200 m/s) | Unknown, NS, SS, RS | RotD50 | Greece |
| Boore et al. (2021) without bias [13] | $M_w$/4.0–8.0 | $R_{jb}$/1–300 km | PGA, PGV, $S_a$ (T = 0.01–10.0 s) | $V_{S30}$-based (150–1200 m/s) | Unknown, NS, SS, RS | RotD50 | Greece |

[*] NS: normal slip, SS: strike slip, RS: reverse slip or thrust.

The final collection procedure includes seven (Table 1) local and regional developed GMPEs based on available strong motion data at that time in Greece, Turkey, and worldwide. Since the majority of the regional/global GMPEs provide estimates of peak ground parameters (e.g., Peak Ground Acceleration (PGA) or/and Peak Ground Velocity (PGV)), the evaluation was initially limited to both intensity measures of ground motion, namely PGA and PGV. The evaluation of each model was made against strong motion data recorded in Greece and Turkey with corresponding earthquakes in their CBA. More specifically, 240 accelerometer recordings were used, provided by three (3) CBA earthquakes which occurred in the period 2017 to 2020 (the Lesvos–Karaburun-Izmir in 2017 [M6.3], the Bodrum–Kos in 2017 [M6.6], and the Samos in 2020 [M7.0]). These recordings were not part of any data set used in generating the GMPEs tested in this study. For testing and ranking, three methods were used, namely, the normalized residuals [14], the likelihood (LH) [15–17], and the log-likelihood (LLH) [18].

(a)     Normalized Residuals Method

The comparison of GMPEs with the observed values of ground motions not included in their production serve several purposes. The main purpose is to realize the range in which the GMPE of interest is correlated with the local properties of source, path, and site factors of ground motion. When modelling both epistemic and aleatory variability, each GMPE is considered in the form of a probability lognormal distribution, as determined by Equation (1):

$$\log y_{ij} = \mu\left(m_i,\, r_{ij},\, \boldsymbol{p_{ij}}\right) + Z_{T,i,j}\, \sigma_T, \tag{1}$$

where the $y_{ij}$ represents the ground motion recorded at location $j$ due to an event $i$, the term $\mu(m_i, r_{ij}, \boldsymbol{p_{ij}})$ represents the expected ground motion from an earthquake of magnitude $m_i$, recorded at distance $r_{ij}$, and finally the term $\boldsymbol{p_{ij}}$ corresponds to other model parameters (e.g., site amplification, fault type, or any other). The total uncertainty, denoted by $Z_{T,i,j}$, is modelled as a normal distribution with a mean zero and a total standard deviation equal to $\sigma_T$. Therefore, $Z_{T,i,j}$ is the total normalized residual of the $j$th recording from the $i$th earthquake event.

When calculating the normalized residual, the term $y_{ij}$ is the recorded ground motion, $\mu(m_i, r_{ij}, \boldsymbol{p_{ij}})$ is the mean estimate of the GMPE, and $\sigma_T$ is the total standard deviation of the GMPE. From the above, it follows that:

$$Z_{T,i,j} = \frac{\log y_{ij} - \mu\left(m_i,\, r_{ij},\, \boldsymbol{p_{ij}}\right)}{\sigma_T}, \tag{2}$$

A GMPE is considered as a good fit to the recorded data if its normalized residuals closely follow a standard normal distribution, with a mean zero and standard deviation equal to 1.0. Differences in the mean of the residuals may indicate a tendency for a GMPE to over- or underpredict the records, whereas differences in the standard deviation may suggest an over- or underestimation of ground motion variability.

Figure 2a,b present the PGA and PGV normalized residual distribution, respectively, for the recording values and the GMPEs considered. In these figures, the model of Boore et al. (2021) [7], both with and without bias, exhibits a distribution of residuals very close to a standard normal distribution, thus indicating a good fit with the data. This was expected, since Boore et al. (2021)'s [7] model was derived from strong motion data specifically of the broader Aegean area that covers the CBAs as well. The models of Chiou and Youngs (2014) [7], Chousianitis et al. (2018) [7], and Akkar et al. (2014) [6] also present a good fit to the recorded data for both PGA and PGV. For the rest of the GMPEs, Kotha et al. (2020) [6] and Abrahamson et al. (2014) [6] show a satisfactory fitting to the observed data. After considering PGA and PGV of equal importance as intensity measures, the corresponding residual values can be combined, so that a unique residual-based ranking and weighting scheme can be obtained, as shown in Table 2.

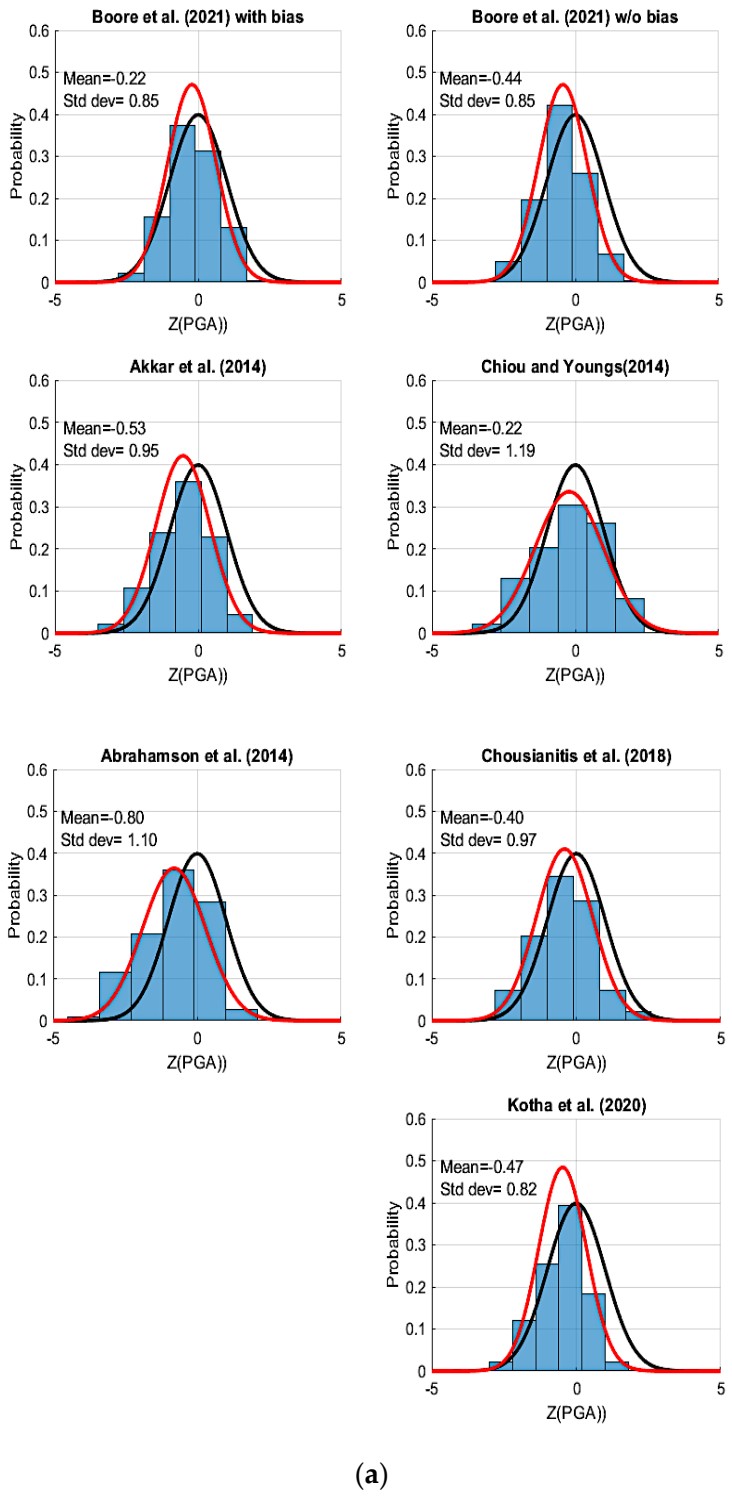

(**a**)

**Figure 2.** *Cont.*

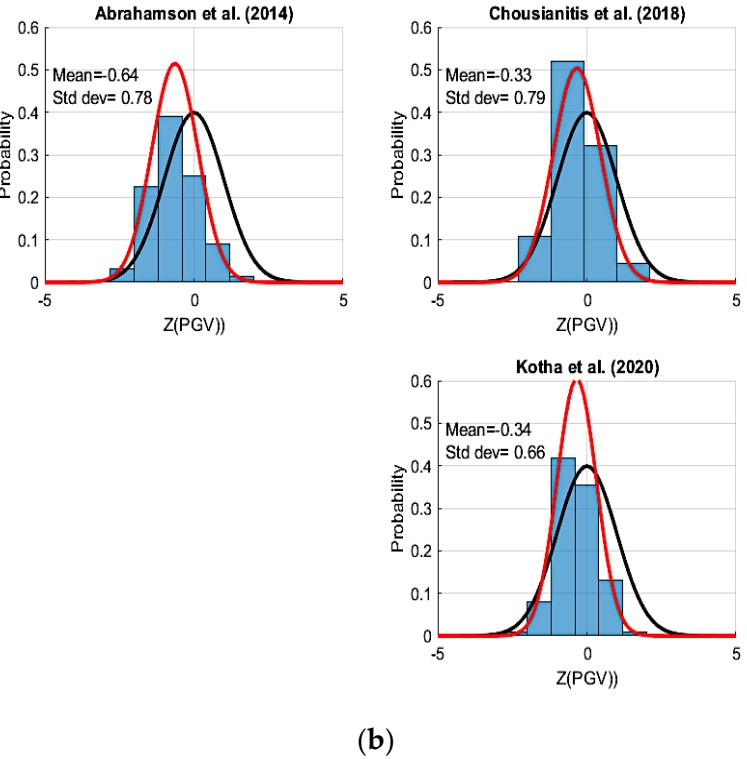

**(b)**

**Figure 2.** (**a**) PGA residual distribution for the records set and the GMPEs considered; (**b**) PGV residual distribution for the records set and the GMPEs considered. The red lines indicate the probability density function fit to the data, whilst the black lines correspond to the standard normal density function.

**Table 2.** Ranking of selected GMPEs based on combined PGA and PGV residuals.

| Ranking | GMPE | Mean Norm. Res. (PGA-GV) | Std. Dev. (PGA-PGV) | Z (PGA-PGV) |
|---|---|---|---|---|
| 1 | Boore et al. (2021) [13] with bias | 0.134 | 0.861 | 0.273 |
| 2 | Chiou and Youngs (2014) [9] | −0.230 | 1.082 | 0.313 |
| 3 | Chousianitis et al. (2018) [11] | −0.286 | 0.861 | 0.424 |
| 4 | Akkar et al. (2014) [8] | −0.369 | 0.887 | 0.483 |
| 5 | Boore et al. (2021) [13] w/o bias | −0.562 | 0.932 | 0.630 |
| 6 | Kotha et al. (2020) [12] | −0.407 | 0.749 | 0.658 |
| 7 | Abrahamson et al. (2014) [10] | −0.720 | 0.951 | 0.769 |

(b) Log-Likelihood method

The simple likelihood analysis (Scherbaum et al., 2004) [17] may not be sufficient, since it may be biased by the sample size and/or subjective view regarding the criteria used to assess the fit of a model. One possible method to measure quantitatively and objectively the goodness-of-fit of GMPEs for a data set is using information theory and the LLH approach, a data-driven evaluation method, proposed by Scherbaum et al. (2009) [18]. The LLH implements an information theoretic approach for the selection and ranking of GMPEs.

The method derived a ranking criterion from the Kullback–Leibler (KL) divergence, which denotes information loss when a model g, defined as a distribution given by a GMPE, is used to approximate a reference model, f, of a data-generation process (i.e., nature). According to Scherbaum et al. (2009) [18], the LLH is finally represented by the negative average sample log-likelihood, according to Equation (3), and is a measure of the distance between a model and the data generation process.

A small LLH indicates that the candidate model is close to the process which has generated the data, while a large LLH corresponds to a model that is less likely to have generated the data.

$$LLH\left(g, \vec{x}\right) := -\frac{1}{N} \sum_{i=1}^{N} log_2(g(x_i)), \tag{3}$$

where $N$ is the number of observations. Scherbaum et al. (2009) [18] provided a potential means of deriving weights for each model within a suite of models, according to Equation (4). In Equation (4), $K$ is the number of GMPEs:

$$w_l = \frac{2^{-LLH}}{\sum_{k=1}^{K} 2^{-LLH}}, \tag{4}$$

The GMPEs used were from Boore et al. (2021) [7] with and without bias, and from Chousianitis et al. (2018) [7], derived from strong motion data of earthquakes in the broader Aegean area. Moreover, the models of Chiou and Youngs (2014) [7], Akkar et al. (2014) [6], Kotha et al. (2020) [6], and Abrahamson et al. (2014) [6] using regional and worldwide observed data were also involved in this analysis. In Probabilistic Seismic Hazard Analysis (PSHA), different GMPEs can be used for different intensity measures; however, this makes the analysis more complex. Considering PGA and PGV to be of equal importance as intensity measures, the corresponding LLH values can be combined, so that a unique LLH-based ranking and weighting scheme can obtained, as shown in Table 3.

**Table 3.** Ranking of selected GMPEs based on combined LLH for PGA and PGV.

| Ranking | GMPE | LLH |
|:---:|:---:|:---:|
| 1 | Chousianitis et al. (2018) [11] | 0.160 |
| 2 | Boore et al. (2021) [13] w/o bias | 0.910 |
| 3 | Boore et al. (2021) [13] w bias | 0.930 |
| 4 | Chiou and Youngs (2014) [9] | 0.932 |
| 5 | Kotha et al. (2020) [12] | 0.971 |
| 6 | Akkar et al. (2014) [8] | 1.035 |
| 7 | Abrahamson et al. (2014) [10] | 1.167 |

(c) Final Selection of GMPEs for Greece–Turkey CBA

The methods of evaluation that were implemented in previous sections constitute an objective way of ranking the pre-selected GMPEs against strong motion data in the Greece–Turkey CBA. As has been observed, different evaluation approaches led to different rankings. Therefore, the final selection of a suite of GMPEs for PSHA and the corresponding weighting resulted from a combination of the rankings presented in Tables 2 and 3. The weighting factors of the final GMPEs was made separately for each evaluation method. Equation (4) applies the LLH approach, whereas for the residual-based approach, the weighting factors are computed in a similar way, according to Equation (5):

$$w_l = \frac{e^{Z*}}{\sum_{k=1}^{K} e^{Z*}} \tag{5}$$

with $Z_* = Z(\text{PGA-PGV})$ as given in Table 2. The weighting factors for the suite of GMPEs are presented in Table 4. The first three provide a weight sum of unity and can be considered as the finally selected GMPEs.

**Table 4.** Final weights of selected GMPEs based on both evaluation approaches.

| A/A | GMPE | $w_l$-LLH | $w_l$-Residuals | Final $w_l$ |
|-----|------|-----------|-----------------|-------------|
| 1 | Boore et al. (2021) [13] with bias | 0.160 | 0.281 | 0.346 |
| 2 | Chiou and Youngs (2014) [9] | 0.160 | 0.270 | 0.337 |
| 3 | Boore et al. (2021) [13] w/o bias | 0.162 | 0.241 | 0.317 |
| 4 | Chousianitis et al. (2018) [11] | 0.272 | 0.121 | - |

## 3. GMPE Selection and Ranking for the Western Black Sea CBA—Romania, Moldova, and Bulgaria

The Western Black Sea CBA of Romania, Moldova, and Bulgaria is affected not only by earthquakes occurring at shallow depths but also at intermediate depths within the Vrancea seismogenic source. Seismic activity in the crustal domain in Romania is rather low and no records of strong/damaging earthquakes are available. Analysis of the observed macroseismic intensities, including the historical part of the catalog, indicates values of VI-VII MSK macroseismic intensity in the examined area of the Moesian Platform and even VIII MSK (in 1981) in the North Dobrogea seismic area. In contrast, the Vrancea intermediate-depth source (VRI) frequently generates large magnitude events with Mw > 7 within a concentrated volume between 60 and 200 km depth, and this seismicity is associated with the current dehydration of an oceanic slab (Ferrand and Manea, 2021 [19] and Craiu et al., 2022 [20]). The impact of VRI events transcends the Romanian national borders and significant damage was also reported in neighboring countries across Western Black Sea CBA, such as VII-VIII MSK during the 1940 Mw 7.7 (Cioflan et al., 2016 [21]; Marmureanu et al., 2016 [22]) and VIII MSK in Northern Bulgaria for the 1802 Mw 7.9 earthquake (Constantin et al., 2011 [23]).

### 3.1. Available Ground Motion Data

The selected engineering parameters of the VRI records are from the dataset developed by Manea et al. (2022) [24]. It contains 421 post-1977 events with Mw ranging from 4 to 7.4 and depths between 60 and 170 km. For crustal seismicity, the selected engineering parameters were obtained from Manea et al. (2017)'s [25] dataset and its update by Cioflan and Manea (2021) [26], and comprises events with magnitudes between 3.5 to 5.6 and maximum depth up to 60 km.

Earthquake parameters used in the test performed for the selected GMPEs were taken from ROMPLUS (INCDFP, 2021) [27], BIGSEES Project (2017) [28], and EMSC for 2017–2020. As the selected GMPEs include the style of faulting (SoF) in their functional forms, complex source information from INCDFP source mechanisms available at http://atlas2.infp.ro/~rt/fms/ (accessed on 28 February 2024), REFMC (Radulian et al., 2019) [29], Craiu et al. (2016; 2017) [30,31] were used. For the earthquakes that occurred in Bulgaria and were recorded by Romanian stations (2003–2019), without available focal mechanisms, the SoF was taken according to the nearest fault characteristics reported in the SHARE model (Woessner et al., 2015) [4]. All these events across the Western Black Sea CBA were recorded by 204 seismic stations of the Romanian National Seismic Network (Available online: https://doi.org/10.1007/978-3-319-14328-6_9, accessed on 28 February 2024) and temporarily deployed within national/international projects of INCDFP since 1985. The database contains also selected records from nine seismic stations deployed within the framework of the DACEA CBC 2011-13 project (Available online: http://quakeinfo.infp.ro/en/, accessed on 28 February 2024) now integrated in the seismic network of the Republic of Bulgaria (Available online: https://doi.org/10.7914/SN/BS, accessed on 28 February 2024) and five stations from the Republic of Moldova (Available online: https://doi.org/10.7914/SN/MD, accessed on 28 February 2024). The site parameters at all the stations were extracted from Manea et al. (2019; 2022) [24,32] and Coman et al. (2020) [33].

## 3.2. Selection of the GMPEs for Intermediate-Depth Vrancea Source

Considering the complex seismotectonic environment of Romania, the ShakeMaps to be used as an input for REDAS should be either generated for the intermediate seismic source of Vrancea or/and for crustal seismicity generated mainly in the western side of the Black Sea. Evaluation of the up-to0date GMPEs for intermediate-depth events (Cioflan et al., 2020) [34] is based on the selection of models from previous seismic hazard analyses, currently used in the Romanian ShakeMaps and recent works (e.g., Douglas 2021) [2]. Several models were considered as good candidates for testing in terms of PGA, PGV, and SA; among them were Atkinson and Boore (2003) [35], Garcia et al. (2005) [36], Lin and Lee (2008) [37], Sokolov et al. (2008) [38], Vacareanu et al. (2015) [39], and Abrahamson et al. (2016) [40]. The main characteristics of the tested GMPEs are presented in Table 5.

**Table 5.** Selected GMPEs for the Vrancea intermediate depth seismogenic area.

| GMPE | Intensity Measures | $M_w$ Range | Distance Range [km] |
|---|---|---|---|
| Abrahamson et al., 2016 (Aetal16) [40] | PGA, SA (<10 s) | 5–7.9 | <300 |
| Atkinson and Boore 2003 (AB03) [35] | PGA, SA (<4 s) | 5–8.3 | <300 |
| Garcia et al., 2005 (Getal05) [36] | PGA, SA (<5 s), PGV | 5.2–7.4 | <400 |
| Lin and Lee 2008 (LL08) [37] | PGA, SA (<5 s) | 5.3–8.4 | <630 |
| Sokolov et al., 2008 (Setal08) [38] | PGA, SA (<3 s), PGV | 6.3–7.4 | <300 |
| Vacareanu et al., 2015 (Vetal15) [39] | PGA, SA (<3.5 s), PGV | 5.1–8 | <400 |

## 3.3. Selection of the GMPEs for Crustal Seismicity

Due to the limited number of recorded data for crustal events in the Western Black Sea CBA, no GMPEs were recently developed in terms of PGA, PGV, or spectral acceleration (SA). For this reason, six internationally recognized GMPEs were selected (Table 6; Manea et al., 2017) [25] to be tested for future implementation in ShakeMaps generation. We chose four developed regional ground motion models based on European and Middle East data and two global ones based on the NGA2 database, whose main characteristics are shown in Table 6.

**Table 6.** Selected GMPEs for crustal seismicity.

| GMPE | Intensity Measures | $M_w$ Range | Distance Range [km] |
|---|---|---|---|
| Akkar et al. (2014) [6] | PGA, SA (<4 s) | 4–7.6 | 0–200 |
| Bindi et al. (2014) [41] | PGA, SA (<3 s) | 4–7.6 | 0–300 |
| Boore et al. (2014) [42] | PGA, SA (<10 s) | 3–7.9 | 0–400 |
| Cauzzi et al. (2014) [43] | PGA, SA (<10 s) | 4.5–7.9 | 0–150 |
| Chiou and Youngs (2014) [7] | PGA, SA (<10 s) | 3.5–8.5 | 0–300 |
| Kale et al. (2015) [44] | PGA, SA ($\leq$4 s) | 4–8 | 0–200 |

## 3.4. Final Selections of the GMPEs

Two traditional tests were performed to evaluate the performance of the selected intermediate-depth and crustal GMPEs using the available data: one is related to the computation of the relative residuals and another one is related to statistical goodness-of-fit parameters. In the first test, relative errors ($\log(PGA_{obs}) - \log(PGA_{GMPE})$) were computed for four selected GMPEs, in terms of PGA, PGV, and spectral accelerations, SA (computed up to 10 s with a step of 0.1 s). Some examples of the computed residuals distribution for representative intermediate-depth and crustal GMPEs as a function of hypocentral distance and their probability density function are shown in Figures 3 and 4.

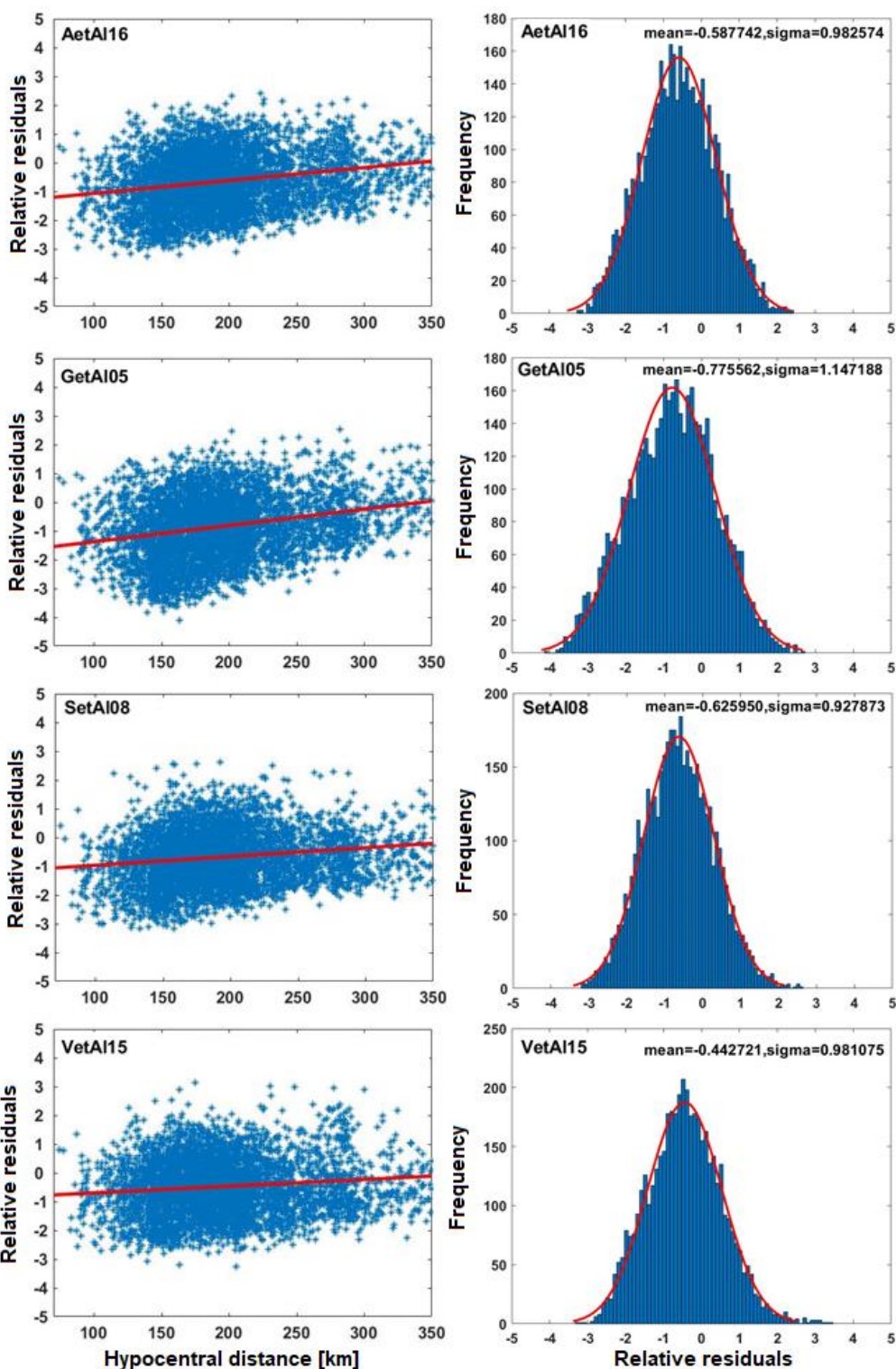

**Figure 3.** An example of probability density function of residuals for the four in-slab GMPEs of Table 5. Where: Aetal16-Abrahamson et al., 2016 [40]; Getal05-Garcia et al., 2005 [36]; Setal08-Sokolov et al., 2008 [38]; Vetal15-Vacareanu et al., 2015 [39].

In the second test, the selected models were ranked using two statistical methods: the likelihood (LH; Scherbaum et al., 2004 [17]) and log-likelihood (LLH; Scherbaum et al., 2009 [18]). These methods were applied for the GMPEs adopted in the Greece–Turkey CBA and they are described in Section 2(b) herein. Our final weighted combination of the GMPEs, which are used in REDAS ShakeMap, is provided for PGA and PGV in Tables 7 and 8.

**Table 7.** Results of LH and LLH tests for Vrancea intermediate-depth seismicity.

| GMPE | PGA | | PGV | | Final Weights |
|---|---|---|---|---|---|
| | LH | LLH | LH | LLH | |
| Abrahamson et al. (2016) [40] | 0.38 | 0.71 | * | * | - |
| Atkinson and Boore. (2003) [35] | 0.44 | −0.25 | * | * | - |
| Garcia et al. (2005) [36] | 0.33 | 0.83 | 0.3 | 0.72 | 0.308 |
| Lin and Lee (2008) [37] | 0.22 | 1.14 | * | * | - |
| Sokolov et al. (2008) [38] | 0.4 | 0.67 | 0.38 | 0.53 | 0.314 |
| Vacareanu et al. (2015) [39] | 0.45 | 0.47 | 0.26 | 1.06 | 0.378 |

(*): Not available GMPEs for PGV.

**Table 8.** Ranking results for crustal earthquake GMPEs.

| GMPE | PGA | | PGV | | Final Weights |
|---|---|---|---|---|---|
| | LH | LLH | LH | LLH | |
| Akkar et al. (2014) [6] | 0.38 | −0.39 | 0.49 | 0.13 | - |
| Bindi et al. (2014) [41] | 0.32 | −0.85 | 0.49 | −0.33 | - |
| Boore et al. (2014) [42] | 0.30 | −0.86 | 0.50 | −0.19 | 0.214 |
| Cauzzi et al. (2014) [43] | 0.41 | −0.29 | 0.28 | 0.96 | 0.297 |
| Chiou and Youngs (2014) [7] | 0.33 | −0.38 | 0.42 | 0.03 | 0.222 |
| Kale et al. (2015) [44] | 0.44 | −0.47 | 0.49 | 0.22 | 0.267 |

For Vrancea intermediate-depth events, the GMPEs are ranked at PGA as follows: Sokolov et al., 2008 [38], Atkinson and Boore 2003 [35], Vacareanu et al., 2015 [39], Garcia et al. 2005 [36]. The Getal05 and Setal08 models have the highest capability to predict PGV while the others have the lowest grades or are even outside of the acceptance range. Discussing the new tools developed for assessing seismic hazard, Solakov et al. (2020) [45] concluded that Vetal15 can be used for the PSHA in Bulgaria in the case of Mw > 6.2 intermediate-depth events. Recently, a new GMPE for Vrancea intermediate-depth events has been developed (Manea et al., 2022) [24] using the data described in Section 3.1. The model is region-specific (only Vrancea records have been used) and is the most appropriate to describe seismic motion generated in the Western Black Sea CBA, as it is based on data from Romania, Moldavia, and Northen Bulgaria. The GMPE was introduced as a proxy measure for the site response, the fundamental frequency of resonance raising new challenges in its implementation for such a wide area.

In the case of crustal earthquakes, the statistical testing is presented in Table 8 and shows that Kale et al. (2015) [44] performed the best for PGA and PGV. In the long period (T > 3 s) of the ground motion, the performance of all GMPEs (except for Boore et al., 2014) [42] dramatically drops. Average LLH and corresponding weights were computed only for those models whose prediction(s) remain in the acceptable range according to the criteria of Scherbaum et al. (2004) [17].

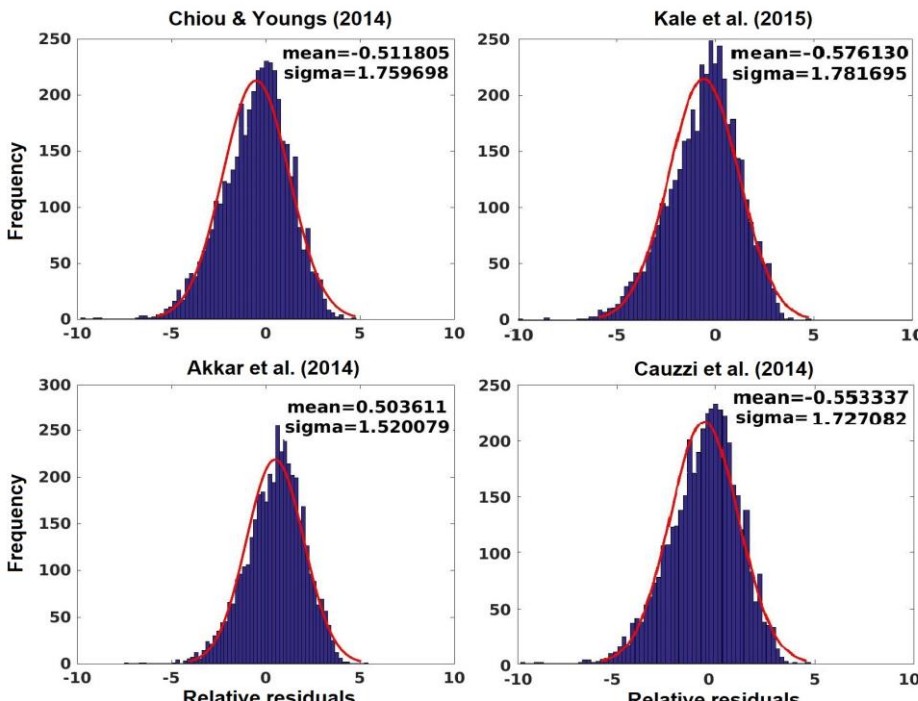

**Figure 4.** An example of residual histograms for four crustal GMPEs of Table 5: Chiou and Youngs (2014) [7], Kale et al. (2015) [44], Akkar et al. (2014) [6], and Cauzzi et al. (2014) [43].

When interpreting these results, one should not neglect the fact that the above GMPEs were tested well below their magnitude limits/range.

## 4. Discussion and Conclusions

In this study, a testing and ranking procedure has been performed to evaluate and select the most appropriate GMPEs for implementation in the REDA System, deployed for the Black Sea Basin area, based on a sufficiently large number of published Ground Motion Prediction Equations (GMPEs). Regarding the cross-border area of Greece and Turkey, three GMPEs were proposed as most appropriate and representative of the regional active shallow crustal seismotectonic setting (Table 4).

One can see (Table 8) that the proposed models (GMPEs) for the Western side of the Black Sea have rather similar performance when considering results of the tests for PGA and PGV. Finally, we proposed a combination of GMPEs with weights computed, also considering their behavior at longer periods to be used for shallow crustal events (see Table 8) across the Western Black Sea territory (Romania, Republic of Moldova, and Republic of Bulgaria). For intermediate-depth events, we recommend the use of the three GMPEs with weights presented in Table 7 or the use of a single model, Manea et al. (2022) [24]. The GMPEs selected for use by the REDAS can be also used in probabilistic seismic hazard assessment in the Black Sea region with the aim of harmonizing seismic risk assessment and mitigation actions.

Regarding the actual accelerometer data input in the REDA System, it must be mentioned that each network operator has their own regulations of operation to achieve in real time, and data transmission to their own computer center. Exact ground motion intensity measures to be used by the REDAS (i.e., PGA, PGV, PSA(T) for various natural periods, T) have been jointly decided by all Partners. All these intensity measures are automatically uploaded to a folder in the "Cloud", which is accessed by the REDAS functionality. The entire procedure is analytically discussed by Papatheodorou et al. (2023) [1].

An example of the GMPEs' implementation of the Samos, 30 October 2020, mainshock (M7.0) REDAS scenario in the Greece–Turkey cross-border area is shown in Figure 5. Estimated ground motion in combination with respective fragility curves of structures

and infrastructure in the CBA, could lead to timely and efficient joint actions of civil protection authorities of both countries to mitigate destructive earthquake consequences. In fact, based on the GMPEs proposed in this study, the scenario generated by the REDA System for the earthquake of Samos, 30 October 2020, M7.0 (Figure 5), is generally in good agreement with the observed macroseismic results due to the mainshock in the broader affected area (Available online: https://www.itsak.gr/uploads/news/earthquake_reports/ EQ_Samos_20201030_report_v3.pdf, accessed on 28 February 2024). However, locally observed high-intensity measures cannot be captured by the System, since specific site amplification factors and vulnerability of actual constructions are not specifically described in the System's database. Such an improvement could be achieved in a future step.

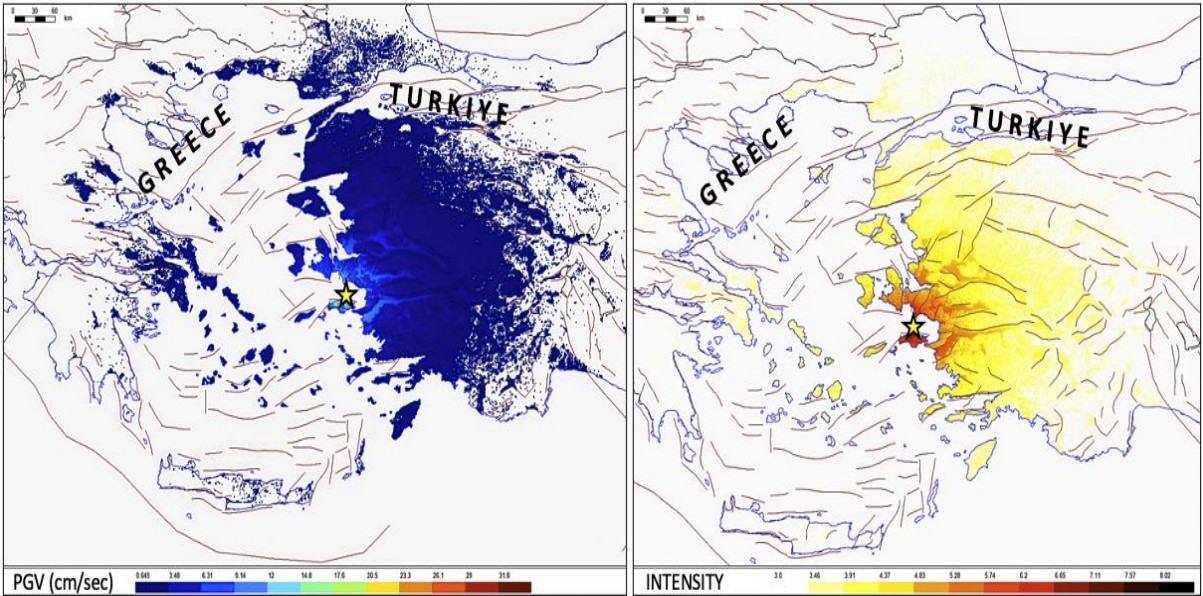

**Figure 5.** Results of the Rapid Earthquake Damage Assessment System (REDAS) in the Greece–Turkey cross border area from the Samos 2020 mainshock (M7.0, yellow star), based on the selected harmonized GPMEs, left: Peak Ground Velocity, right: Macroseismic Intensity.

The REDAS is a dynamic system (Papatheodorou et al., 2023) [1] that can adopt new information in the future relative to seismic risk management and its mitigation. It must also be stated that additional data to be acquired in the future within the broader cross-border areas of the Black Sea Basin would contribute to defining or/and testing new GMMs compatible with the seismotectonic environment of the examined territory. In fact, there has recently been an additional and ever-increasing number of GMMs developed in the broader Black Sea Basin that could have been considered in our analyses (among others; Zafarani et al. (2018) [46], Farajpour et al. (2020) [47], Darzi et al. (2020) [48]). But the lack of intensity measures to be examined (like PGV in GMMs of [46,47]) drove us to their exclusion from further analyses. As for the GMM of [48], its limitation in epicentral distance, R ≤ 200 km, drove us to exclude it as well. However, it should be mentioned that including all recent advancements in GMMs may enhance the accuracy and reliability of seismic hazard assessment in the study area. In future study, by relaxing several strict criteria in favor of the most recent GMM inclusion, we could gain higher reliability and lower uncertainly in our results. This could be a never-ending process by adapting suitable GMMs and improving seismic hazard and risk assessment in the Black Sea Basin.

**Author Contributions:** Conceptualization, N.T.; methodology, N.T., B.M., C.O.C., E.F.M. and D.T.-D.; software, D.S., C.O.C., E.F.M. and D.T.-D.; validation, B.M., D.S., C.Z., S.O.A., C.O.C. and E.F.M.; formal analysis, B.M. and C.Z.; investigation, E.F.M.; data curation, C.Z., S.O.A. and C.O.C.; writing— original draft preparation, N.T.; writing—review and editing, N.T., B.M., D.S., C.Z., S.O.A. and C.O.C.;

supervision, N.T.; project administration and funding acquisition, N.T. All authors have read and agreed to the published version of the manuscript.

**Funding:** The REDACt project (BSB-966) has been financially supported by the Black Sea Basin joint Operational Programme 2014-20; Grant contract no. <MLPDA 88712/26.06.2020>.

**Data Availability Statement:** Project produced data can be made available upon request from thesection "Contact Us" of the project Website: https://www.redact-project.eu/ (accessed on 28 February 2024).

**Acknowledgments:** This work has been supported by the "Rapid Earthquake Damage Assessment Consortium": REDACt project, eMS Code: BSB966, funded by the Black Sea Basin Joint Operational Programme 2014-20 co-financed by the European Union through the European Neighbourhood Instrument and by the participating countries: Armenia, Bulgaria, Georgia, Greece, Republic of Moldova, Romania, Turkey, and Ukraine.

**Conflicts of Interest:** The authors declare no conflict of interest.

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
