# Peer review of "Rapid Earthquake Damage Assessment System in the Black Sea Basin: Selection/Adoption of Ground Motion Prediction Equations with Emphasis in the Cross-Border Areas"

_2624-795X, doi:10.3390/geohazards5010013_

Round 1
Reviewer 1 Report
Comments and Suggestions for Authors
The manuscript "Rapid Earthquake Damage Assessment System in the Black Sea Basin: Selection/Adoption of GMPEs with Emphasis in the 3 Cross-Border Areas " focuses on assessing Ground Motion Models (GMMs) for seismic activity in the Black Sea region, particularly in cross-border areas. While the research provides valuable insights, some parts need attention to make the results more complete and reasonable.
1. The study didn't clearly highlight the differences between the Rapid Earthquake Damage Assessment System in the Black Sea and older versions. It would be beneficial to emphasize these distinctions, especially since the study included the old ranking methods (LH and LLH) and old version GMMs.
2. A significant concern for this study is the exclusion of recent advancements in Ground Motion Models (GMMs). The study should consider incorporating newer models developed for shallow earthquakes, such as those presented in works like Zafarani et al. (2018), Farajpour et al. (2020), and Darzi et al. (2020) based on dense and new strong-motion database for shallow-earthquakes in the region. Including the advancements will enhance the study's accuracy and relevance to the current state of seismic hazard research.
3. References for the methods used, such as LH and LLH, are crucial for other researchers to understand and explore the study further. Providing detailed references for these methods would strengthen the research's credibility and encourage future discussions and comparisons.
4. The study acknowledges limitations in the LH method but falls short in explaining why more recent and advanced data-driven evaluation methods, like Mac et al. (2017) and Kowsari et al. (2019), were not employed. Including these methodologies could enhance the study's accuracy and relevance.
5. References ((a) Normalized Residuals Method) should be included to allow readers to trace the theoretical foundations of the mathematical models presented.
6. The references section is incomplete, with missing papers like Kale et al. (2015). Please make sure a comprehensive list of references is included in the study.
Reviewer 2 Report
Comments and Suggestions for Authors
The paper of Nikolaos et al. proposed the investigation of several models to predict Ground motion induced by earthquakes in country border areas. In particular, the authors focused in the area around the Black Sea basin. This work is very important and I would recommend its publication as underlined by the authors at the border between countries the precision of regional (National) models is affected. The study is consequently precious also to mitigate future seismic hazard risk in the investigated area. The authors in particular examined 7 most updated Ground Motion Prediction Equations (GMPEs) testing at each border which one performed better compared with real data of a strong motion network not used to construct the models. I suggest so to accept the paper after some revision of a couple of major questions and some minors points and a few typos.
Major questions:
· For the evaluation of the Gaussian distribution of the difference between PGA from one of the 7 models under evaluation and the one from ground motion (shown in Figure 2) I think you can introduce a statistical test (or parameter) such as Kolmogorov–Smirnov test. Can you add, please?
· Can you add in conclusion a comment about how your method could be further improved in the future?
Minors points:
· Line 30. Please define all the acronyms even if already defined in the Abstract (GMPE)
· Lines 37-40. I suggest explaining a bit the reason for the preference for a country model (for example: a better regional model of the ground and/or a more dense seismic network)
· Line 45. This sentence is not so clear. I suggest replacing “territory, the adoption of the most appropriate GMPEs for the seismotectonic setting is attempted”
· Lines 75-78. I am not arguing to restrict the analysis to the most recent GMPSs, but I would like you to add a bit more details about the exclusion criteria (for example what are the ones you cited of Cotton et al., 2006?)
· Line 79. Sorry, is it “evaluation process” the most proper expression? What about the “collection process”?
· Line 82. Please insert the acronyms in the first time that appear: Peak Ground Velocity and Acceleration (PGV and PGA).
· Line 84. What do you mean for “causative”? I don’t understand sorry
· Line 95. I think it’s better say “serve for several purposes” (add for)
· Line 205. Please check the alphabet of “and” (Latin not Greek)
· Line 286. I suggest anticipating (T>3s) after “long period”.
· Lines 294-297. I would suggest shifting the sentence “based on ... (GMPEs)” after “Black Sea Basin area”, i.e., at the end of sentence. Futhermore please add a comma after “In this study, ”
· Line 307-309. The font is different.
Comments on the Quality of English Language
English is fine from my point of view. A few suggestions to improve the fluency are included in the previous list.
Reviewer 3 Report
Comments and Suggestions for Authors
The manuscript proposed GMPEs to 20 feed the REDA System, which may assure effective estimation of ShakeMaps and - in combination with 21 the appropriate vulnerability curves - reliable near real time damage assessment in the cross border 22 earthquake affected areas.The study provided valuable insights into the damage assessment processes that is using the Ground Motion Prediction Equations and focused on understanding the relationship between Ground Motion Prediction Equations and Rapid Earthquake Damage Assessment System in the Cross Border Areas, which is a topic of significant interest in the field of earthquake research. There are some suggestions for improvement:
(1) The author mentioned in the article that earthquake damage assessment can be quickly achieved. What is the efficiency and accuracy of the assessment? And what is the input?
(2) What are the characteristics of the Cross Order Area (CBA) of the broker Black Basin territory and what is the current situation? I suggest the author explain it in the text? Another issue is that the data obtained by the author are all existing data from other literature. How can we determine the accuracy of the data?
(3) What are the standards for monitoring data in different regions? How did the author solve this problem? It is recommended that the author provide a detailed description in the article.
(4)The author elaborated on the results obtained by their method in the article, but did not compare the existing results with the results obtained by the author's proposed method. It is suggested to supplement them.
(5) I suggest the author carefully check for formatting errors in the text, such as line 307.
(6)The format of references needs to be adjusted according to the requirements of the journal.
Comments on the Quality of English LanguageMinor editing of English language required
Reviewer 4 Report
Comments and Suggestions for Authors
This is a well-written article with clear and understandable language and a proper bibliography of the field. All the figures (graphs and maps) and tables represent accurately the results, The subject of the article has an important impact on the field and the study design was appropriate for answering all the questions.
The "Abstract" and "Discussion and Conclusions" sections were clearly written, giving accurate information on the research and the results, without spin for the reviewer and the reader as long as it gets published and accessed to a broader audience.
Therefore, I recommend this research paper be published in its present form.
Kind regards
Round 2
Reviewer 1 Report
Comments and Suggestions for Authors
Thanks for sending the revised version.
Author Response
The Reviewer 1 states "Thanks for sending the revised version" and he has no more comments/suggestions on the manuscript. I guess that he is covered by the revised version of the manuscript and corresponding replies to his/her remarks made in the first round. After that I would like to thank the Reviewer 1 for his comments/suggestions made in round one that substantially improved the manuscript.
Reviewer 2 Report
Comments and Suggestions for Authors
Dear authors,
Thank you so much for your reply to my questions. I'm fully satisfied with your answers and revised version of the manuscript, which I found improved with respect to the first submission. Your explanation of why it's better not to include a statistical test (S-K) is completely reasonable, and so I agree with you. I thank you to have considered all my points.
I recommend to the editor to accept your paper for publication in Geohazards.
Thank you so much!
Author Response
The Reviewer 2, states that :
"Thank you so much for your reply to my questions. I'm fully satisfied with your answers and revised version of the manuscript, which I found improved with respect to the first submission. Your explanation of why it's better not to include a statistical test (S-K) is completely reasonable, and so I agree with you. I thank you to have considered all my points."
and he/she recommends to the editor acceptance of the paper for publication in Geohazards.
After that, I would like to thank the Reviewer 2 for his/her valuable comments/remarks that have certainly improved the final version of the manuscript.
Reviewer 3 Report
Comments and Suggestions for Authors
All questions have been answered, and I believe it can be published
Comments on the Quality of English LanguageAll questions have been answered, and I believe it can be published
Author Response
The reviewer 3 states that:
"All questions have been answered, and I believe it can be published."
In addition he suggests acceptance of the paper for publication in the Bulletin. After that, I would like to thank the Reviewer 3 for his/her comments/remarks that certainly improved the manuscript before its publication to Geohazards.